# Improving SAM Requires Rethinking its Optimization Formulation

## Abstract

This paper rethinks Sharpness-Aware Minimization (SAM), which is originally formulated as a zero-sum game where the weights of a network and a bounded perturbation try to minimize/maximize, respectively, the same differentiable loss. We argue that SAM should instead be reformulated using the 0-1 loss, as this provides a tighter bound on its generalization gap. As a continuous relaxation, we follow the simple conventional approach where the minimizing (maximizing) player uses an upper bound (lower bound) surrogate to the 0-1 loss. This leads to a novel formulation of SAM as a bilevel optimization problem, dubbed as BiSAM. Through numerical evidence, we show that BiSAM consistently results in improved performance when compared to the original SAM and variants, while enjoying similar computational complexity.

## 1 Introduction

The rise in popularity of Large Language Models (LLMs) has motivated the question of which optimization methods are better suited for their training. Recently, it has been found that *Sharpness-Aware Minimization (SAM)* (Foret et al., 2021) can greatly improve their generalization with almost negligible increase in computational complexity (Bahri et al., 2021). SAM not only improves the performance of LLMs, but supervised learning tasks from computer vision also benefit greatly (Foret et al., 2021; Dosovitskiy et al., 2020). Hence, it is natural to ask whether SAM can be improved further. Indeed, many works have been quick to present modifications of the original SAM algorithm, that improve its speed (Du et al., 2022) or performance (Kwon et al., 2021) in practice.

The motivation behind SAM is to find a parameter $w^\star$ in the so-called *loss landscape*, that achieves a low *loss value* while being *flat* i.e., the loss in its immediate neighborhood should not deviate meaningfully from the value attained at $w^\star$. Such *loss landscape* is usually understood as the one associated with the cross-entropy, or other common differentiable losses, on the *training set*. Indeed, the most popular method for supervised learning with deep neural networks is to use the cross-entropy as a surrogate of the misclassification error, and minimize it through gradient-based methods in an effort to find highly accurate classifiers.

Why is such *flatness* important? The main theoretical advantage is that it promotes good *generalization*, which can be defined as the difference between a performance measure in the training set vs the *testing set*. This important property has been brought forward by the PAC-Bayesian generalization bounds derived in Dziugaite and Roy (2017), confirmed empirically through extensive evaluation in Keskar et al. (2017); Jiang* et al. (2020) and exploited by Foret et al. (2021) to derive the SAM algorithm. Denoting by $L_S$ the average (cross-entropy) loss on the dataset $S$, the original SAM optimization template is stated as the zero-sum game:

$$\min_w \max_{\epsilon:\|\epsilon\|_2\le\rho} L_S(w+\epsilon) + h\left(\|w\|_2^2/\rho\right) \tag{1}$$

where $h : \mathbb{R}_+ \to \mathbb{R}_+$ is some strictly increasing function. This min-max zero-sum formulation is the starting point for all works that try to improve upon the method. For example, Du et al. (2022) introduce modifications over the original SAM algorithm like perturbing only a random subset of weights in each iteration, or optimizing only on a subset of data that is

more sensitive to the sharpness. Alternatively, Kwon et al. (2021) modify the allowed set of perturbations to make SAM scale-invariant. The use of a surrogate loss $L_S$ in Eq. (1), usually the cross-entropy, is a mainstay in such approaches.

Nevertheless, in the context of supervised classification, the usage of the cross-entropy loss in the definition of SAM (1) presents a concerning subtlety. As SAM controls the generalization of the loss, the formulation (1) is in fact looking for a parameter such that the cross-entropy on the testing set does not deviate from the low value of cross-entropy achieved on the training set. Even though this appears beneficial, we should recall that the goal in supervised classification is not to achieve a low value of the cross-entropy, rather, the goal is to enjoy a small *misclassification error rate* on the testing set. This raises the question:

*If our goal is to achieve a better classifier, should we not apply the SAM formulation directly on the misclassification error i.e., the 0-1 loss?*

In order to answer this question, first of all, we need to make sure that the theoretical properties of generalization that motivate SAM (Dziugaite and Roy, 2017) still hold for the so-called *0-1 loss*, which corresponds to the misclassification error of the model. Assuming we answer this in the positive, the question remains whether the formulation in Eq. (1) is amenable to optimization when the loss is chosen as the 0-1 loss.

In this work, we answer the first question affirmatively, that is, we verify that the theoretical results of Dziugaite and Roy (2017) and Foret et al. (2021) apply directly to the 0-1 loss. The statements therein are of statistical nature, and their proofs do not require any *differentiability* or *continuity* assumptions. Hence, applying SAM directly on the misclassification error indeed leads to better generalization of the actual desired performance metric of a classifier. This suggests we should reframe SAM as:

$$\min_w \max_{\epsilon:\|\epsilon\|_2 \leq \rho} L_S^{01}(w + \epsilon) + h\left(\|w\|_2^2/\rho\right) \tag{2}$$

Unfortunately, the 0-1 loss is discontinuous nature and has zero-gradient almost everywhere, which prevents its direct optimization through gradient-based methods. Therefore, even though 0-1 loss-based SAM has better theoretical guarantees, it does not lead to a practical method. This issue is not foreign in supervised classification: precisely, differentiable surrogate losses like the cross-entropy were originally introduced as upper-bounds on the misclassification error, which motivates their minimization as a proxy for obtaining accurate classifiers through algorithms like SGD.

Following this bound-based approach from textbook ML, one might be inclined to simply replace the 0-1 loss with the cross-entropy, going full-circle and arriving at the original SAM formulation Eq. (1). In this work, we posit that such careless modification would suffer from a fundamental flaw: whereas for the model weights $w$, whose goal is one of *minimization*, it makes sense to use an upper bound on the misclassification error e.g., cross-entropy, the same cannot be said about the weight-perturbation $\epsilon$ whose objective is to *maximize* the error. Instead, a sound approach is for the maximization player $\epsilon$ to optimize a *lower bound* on the misclassification error.

Namely, in order to solve Eq. (2) via gradient-based algorithms, we need to relax the objective of the model weights and adversary independently of each other. This leads to a situation where both players end up with a fundamentally different objective function i.e., a non-zero-sum game. In conclusion, in this work we argue that improving SAM requires moving away from the original zero-sum game formulation Eq. (1) and towards a *bilevel optimization* formulation (Bard, 2013), that can effectively capture different surrogate losses for the two players in the 0-1 loss SAM formulation (2).

We summarize **our contributions** as follows:

- We present (15), a novel bilevel optimization (Bard, 2013) formulation of SAM, where instead of solving a min-max zero-sum game between the model parameters $w$ and the perturbation $\epsilon$, each player has a different objective. This formulation appears naturally by applying SAM to the relevant performance metric in supervised learning: the misclassification error, i.e., the so-called 0-1 loss. Our formulation retains the guarantees of good generalization (8).

- We propose *BiSAM* (Algorithm 1), a scalable first-order optimization method to solve our proposed bilevel formulation of SAM. BiSAM is simple to implement and enjoys a similar computational complexity when compared to SAM.
- We present numerical evidence on CIFAR10/CIFAR100 showing that our proposed reformulation and algorithm consistently outperforms SAM across five models, and also see improvement on ImageNet-1K. BiSAM incorporating variants of SAM (ASAM and ESAM) also demonstrates enhancement. We additionally verify that our reformulation remains robust in finetuning and noisy label tasks.

## 2 Preliminaries and Problem Setup

**Notation.** Throughout this work we let $\mathcal{D}$ be an (unknown) distribution over data-label pairs $(x, y)$ where $x \in \mathbb{R}^d$ and $y \in \{1, \dots, K\} = [K]$, $S = \{(x_1, y_1), \dots (x_n, y_n)\}$ is a finite sample drawn from $\mathcal{D}$. $f_w : \mathbb{R}^d \to \mathbb{R}^K$ corresponds to the logits (scores) that are output by a neural network with parameters $w$. For a given *loss function* $\ell$ we denote the *population loss* $L_{\mathcal{D}}(w) = \mathbb{E}_{(x,y)\sim\mathcal{D}}[\ell(f_w(x), y)]$ and the *training set loss* $L_S(w) = \frac{1}{n}\sum_{i=1}^n \ell(f_w(x_i), y_i)$. We denote the cross-entropy loss as $\ell^{\text{ce}}$, and its corresponding population and training set loss as $L_{\mathcal{D}}^{\text{ce}}, L_S^{\text{ce}}$. We denote as $\{A\}$ the indicator function of an event i.e., $\{A\} = 1$ if $A$ is true or $\{A\} = 0$ if $A$ is false. In this way we can write the 0-1 loss as

$$\ell^{01}(f_w(x), y) = \left\{ \arg\max_{j=1,\dots,K} f_w(x)_j \neq y \right\} \tag{3}$$

For the 0-1 loss we denote its corresponding population and training set loss as $L_{\mathcal{D}}^{01}, L_S^{01}$.

Let us start by recalling that the motivation behind SAM is the PAC-Bayesian generalization bound by Dziugaite and Roy (2017), which leads to the following:

**Theorem 1.** *(Foret et al., 2021, (stated informally)) For any $\rho > 0$, with high probability over training set $S$ generated from distribution $\mathcal{D}$:*

$$L_{\mathcal{D}}(w) \leq \max_{\epsilon: \|\epsilon\|_2 \leq \rho} L_S(w + \epsilon) + h\left(\|w\|_2^2/\rho\right) \tag{4}$$

*where $h : \mathbb{R}_+ \to \mathbb{R}_+$ is a strictly increasing function (under some technical conditions on $L_{\mathcal{D}}(w)$).*

This result suggests that by minimizing both sides of Eq. (4), we can minimize the *population loss*:

$$\min_w L_{\mathcal{D}}(w) \leq \min_w \max_{\epsilon: \|\epsilon\|_2 \leq \rho} L_S(w + \epsilon) + h\left(\|w\|_2^2/\rho\right) \tag{5}$$

By choosing the loss $L = L^{\text{ce}}$ as the cross-entropy, or any other common differentiable loss functions, we arrive at the SAM formulation used in practice. However, taking a step back, we stress that the final goal in supervised learning is to minimize the 0-1 loss at the population level i.e., minimize the expected (test set) misclassification error. This is indeed the metric that is reported experimentally and determines which method is the state-of-the-art in classification tasks.

The reason why differentiable losses like the cross-entropy are introduced are two-fold: first, their differentiability allows the use of first-order optimization methods, and second, they provide an upper bound to the 0-1 loss $L^{01}$. Consequently, minimizing an upper bound leads to a decrease in misclassification error, which is the actual goal of the classifier. In summary, the chain of inequalities that motivate the original SAM formulation (Foret et al., 2021) is the following:

$$\min_w L_{\mathcal{D}}^{01}(w) \leq \min_w L_{\mathcal{D}}^{\text{ce}}(w) \leq \min_w \max_{\epsilon: \|\epsilon\|_2 \leq \rho} L_S^{\text{ce}}(w + \epsilon) + h\left(\|w\|_2^2/\rho\right) \tag{6}$$

Given that the PAC-Bayesian bounds motivating SAM are considered the tightest (Dziugaite and Roy, 2017; Lotfi et al., 2022), it seems like the choice of first upper bounding the 0-1 loss with the cross-entropy loss is suboptimal. *Can we apply directly the PAC-Bayesian bound from Dziugaite and Roy (2017) to the 0-1 loss?* Yes we can:

**Remark 1.** *The PAC-Bayesian bounds from Dziugaite and Roy (2017) hold for the 0-1 loss. Furthermore, after inspecting the proof of Theorem 1 in Foret et al. (2021) we conclude that the following bound also applies to the 0-1 loss:*

$$L_{\mathcal{D}}^{01}(w) \leq \max_{\epsilon: \|\epsilon\|_2 \leq \rho} L_S^{01}(w + \epsilon) + h\left(\|w\|_2^2/\rho\right) \tag{7}$$

With this in mind, we decide to delay the introduction of differentiable losses like cross-entropy, and instead start from the inequality:

$$\min_w L_{\mathcal{D}}^{01}(w) \leq \min_w \max_{\epsilon: \|\epsilon\|_2 \leq \rho} L_S^{01}(w + \epsilon) + h\left(\|w\|_2^2/\rho\right) \tag{8}$$

we now turn to the question of how to solve the problem in the right-hand-side of Eq. (8). To make this formulation amenable to first-order optimization, we need to replace the discontinuous 0-1 loss by a neural network. However, due to the min-max formulation in Eq. (8), it would be wrong to simply replace it by the cross-entropy.

While such approach would work for the minimization player (corresponding to the variable $w$), it is not valid for the maximization player (corresponding to the variable $\epsilon$). When the goal is to minimize a given untractable objective, it is sensible to minimize a differentiable upper bound. Analogously, if the goal is to *maximize* an objective like the 0-1 loss, one can instead maximize a *lower bound*. Otherwise, maximizing an upper bound leads to no guarantees whatsoever on the true objective. Precisely, this asymmetry is what will lead to a natural formulation of SAM as a bilevel optimization problem (Bard, 2013).

## 3 Bilevel Sharpness-aware Minimization (BiSAM)

In order to obtain differentiable objectives for the minimization and maximization players in the formulation Eq. (8), our starting point is to decouple the problem as follows:

$$\min_w L^{01}(w + \epsilon^\star) + h\left(\|w\|_2^2/\rho\right), \qquad \text{subject to } \epsilon^\star \in \arg\max_{\epsilon: \|\epsilon\|_2 \leq \rho} L^{01}(w + \epsilon) \tag{9}$$

Up to this point, there has been no modification of the original objective Eq. (8). We first note that for the minimization player $w$, we can minimize a differentiable upper bound (e.g., cross-entropy) instead of the 0-1 loss, leading to the formulation:

$$\min_w L^{ce}(w + \epsilon^\star) + h\left(\|w\|_2^2/\rho\right), \qquad \text{subject to } \epsilon^\star \in \arg\max_{\epsilon: \|\epsilon\|_2 \leq \rho} L^{01}(w + \epsilon) \tag{10}$$

Now, we need only deal with replacing the 0-1 loss in the objective of the perturbation $\epsilon$. Because this corresponds to a maximization problem, we need to derive a *lower bound*.

**Lemma 1.** *Let $\phi(x)$ be a lower bound of the 0-1 step function $\{x > 0\}$. For each $j \in [K]$, let $F_{w+\epsilon}(x_i, y_i)_j = f_{w+\epsilon}(x_i)_j - f_{w+\epsilon}(x_i)_{y_i}$ and let $\mu > 0$. It holds that*

$$L^{01}(w + \epsilon) \geq \frac{1}{n} \sum_{i=1}^{n} \frac{1}{\mu} \log\left(\sum_{j=1}^{K} e^{\mu\phi(F_{w+\epsilon}(x_i,y_i)_j)}\right) - \frac{1}{\mu} \log(K) \tag{11}$$

**Remark 2.** *Choice $\phi(x) = \tanh(x)$ is a valid lower bound (see (17) for further discussion).*

*Proof.* Note that for a training sample $(x_i, y_i)$ we have misclassification error if and only if for some class $j \neq y_i$ the score assigned to class $j$ is larger than the score assigned to $y_i$. Equivalently, $\arg\max_{j\in[K]} f_{w+\epsilon}(x_i)_j \neq y_i$ if and only if $\max_{j=1,\dots K} F_{w+\epsilon}(x_i, y_i)_j > 0$. Thus,

$$
\begin{aligned}
L^{01}(w + \epsilon) &= \frac{1}{n} \sum_{i=1}^{n} \left\{ \arg\max_{j\in[K]} f_{w+\epsilon}(x_i)_j \neq y_i \right\} = \frac{1}{n} \sum_{i=1}^{n} \left\{ \max_{j\in[K]} F_{w+\epsilon}(x_i, y_i)_j > 0 \right\} \\
&= \frac{1}{n} \sum_{i=1}^{n} \max_{j\in[K]} \{F_{w+\epsilon}(x_i, y_i)_j > 0\} \geq \frac{1}{n} \sum_{i=1}^{n} \max_{j\in[K]} \phi\left(F_{w+\epsilon}(x_i, y_i)_j\right)
\end{aligned}
\tag{12}
$$

To end up with a differentiable expression, we need only get rid of the non-differentiable maximum operator over the set $[K] = \{1, \dots, K\}$. To this end we use the well-known bounds

of the log-sum-exp function:

$$\frac{1}{\mu} \log \left( \sum_{i=1}^{K} e^{\mu a_i} \right) \leq \max\{a_1, \ldots, a_K\} + \frac{1}{\mu} \log(K) \tag{13}$$

Using Eq. (13) in Eq. (12) yields the desired bound. □

As a consequence of Lemma 1 we conclude that a valid approach for the maximization player is to solve the differentiable problem in the right-hand-side of the following inequality:

$$\max_{\epsilon:\|\epsilon\|_2 \leq \rho} L^{01}(w + \epsilon) \geq \max_{\epsilon:\|\epsilon\|_2 \leq \rho} \frac{1}{n} \sum_{i=1}^{n} \frac{1}{\mu} \log \left( \sum_{j=1}^{K} e^{\mu \phi(F_{w+\epsilon}(x_i, y_i)_j)} \right) - \frac{1}{\mu} \log(K)$$

$$=: \max_{\epsilon:\|\epsilon\|_2 \leq \rho} Q_{\phi,\mu}(w + \epsilon) \tag{14}$$

Continuing from Eq. (10), we finally arrive at a bilevel and fully differentiable formulation:

$$\min_{w} L^{ce}(w + \epsilon^\star) + h\left(\|w\|_2^2 / \rho\right), \qquad \text{subject to } \epsilon^\star \in \arg\max_{\epsilon:\|\epsilon\|_2 \leq \rho} Q_{\phi,\mu}(w + \epsilon) \tag{15}$$

The above setting forces us to take a slight detour to the general framework of bilevel optimization and its solution concepts. In particular, the nested nature of the problem makes its solution to be notoriously difficult. Therefore, the success of the up-to-date iterative methods relies on a set of quite restrictive assumptions, which do not apply in the complex environment of neural networks (we refer the reader to Ghadimi and Wang (2018); Tarzanagh and Balzano (2022) for more details). In particular, an important feature that needs to be satisfied is that the so-called inner problem should be strongly convex; which here is clearly not the case. Therefore, in order to devise a fast algorithm for the problem in the right-hand-side of (15), some particular modifications should be made. More precisely, we follow the same approach in the original SAM algorithm (Foret et al., 2021), and do a first-order Taylor expansion of $Q_{\phi,\mu}(w + \epsilon)$ with respect to $\epsilon$ around 0. We obtain:

$$\epsilon^\star \in \arg\max_{\epsilon:\|\epsilon\|_2 \leq \rho} Q_{\phi,\mu}(w + \epsilon) \approx \arg\max_{\epsilon:\|\epsilon\|_2 \leq \rho} Q_{\phi,\mu}(w) + \epsilon^\top \nabla_w Q(w)$$

$$= \arg\max_{\epsilon:\|\epsilon\|_2 \leq \rho} \epsilon^\top \nabla_w Q(w) = \rho \frac{\nabla_w Q(w)}{\|\nabla_w Q(w)\|_2} \tag{16}$$

As the function $Q_{\phi,\mu}$ involves a sum over the whole dataset, this makes the computation of the full gradient in Eq. (16) too costly. For scalability, in practice we use stochastic gradients defined on a mini-batch. Our proposed algorithm to solve SAM in the bilevel optimization paradigm (BiSAM), finally takes shape as shown in Algorithm 1.

---

**Algorithm 1** Bilevel SAM (BiSAM)

---

**Input:** Initialization $w_0 \in \mathbb{R}^d$, iterations $T$, batch size $b$, step sizes $\{\eta_t\}_{t=0}^{T-1}$, neighborhood size $\rho > 0$, $\mu > 0$, lower bound $\phi$.

1 **for** $t = 0$ **to** $T - 1$ **do**
2      Sample minibatch $\mathcal{B} = \{(x_1, y_1), \ldots, (x_b, y_b)\}$.
3      Compute the (stochastic) gradient of the perturbation loss $Q_{\phi,\mu}(w_t)$ defined in Eq. (14)
4      Compute perturbation $\epsilon_t = \rho \frac{\nabla_w Q(w)}{\|\nabla_w Q(w)\|}$.
5      Compute gradient $g_t = \nabla_w L_{\mathcal{B}}(w_t + \epsilon_t)$.
6      Update weights $w_{t+1} = w_t - \eta_t g_t$.

---

**On the choice of lower bound $\phi$.** The function $\phi$ plays a crucial role in the objective $Q_{\phi,\mu}$ that defines the perturbation Eq. (16). Although in theory we can use any lower bound for the 0-1 step function $\{x > 0\}$, the choice can affect the performance of the optimization algorithm. As is always the case in Deep Learning, one should be on the look for possible sources of *vanishing/exploding gradients* (Hochreiter et al., 2001).

As shown in Fig. 1, the function

$$\phi(x) = \tanh(\alpha x) \tag{17}$$

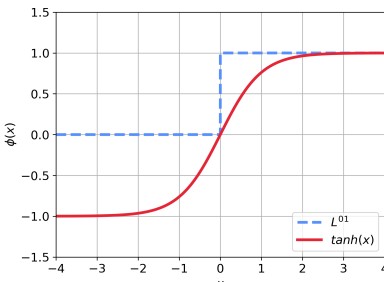 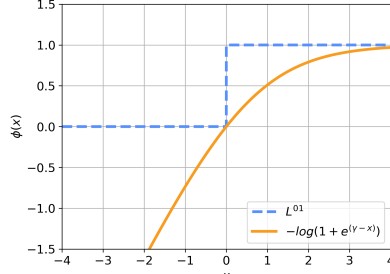

**Figure 1:** *Plot of suggested lower bounds. Left:* $\phi(x) = \tanh(\alpha x)$, *Right:* $\phi(x) = -\log(1 + e^{(\gamma - x)}) + 1$.

seems to be a good lower bound of the 0-1 step function. However, at all points far from zero, the gradient quickly vanishes, which might harm performance. We suggest considering the alternative:

$$\phi(x) = -\log(1 + e^{(\gamma - x)}) + 1 \tag{18}$$

where $\gamma = \log(e - 1)$, also shown in Fig. 1, as it only suffers from a vanishing gradient on large positive values. However, note that having a vanishing gradient in such region is not really an issue: the objective of the perturbation $\epsilon$ is to move points towards the right side of the plot in Fig. 1, where misclassification happens. Hence, if a point stays there due to the vanishing gradient problem, it means it will remain misclassified. In contrast, having vanishing gradients on the left side of the plot in Fig. 1 might mean that the optimization algorithm is unable to move points that are correctly classified towards the misclassification region, therefore the adversary would fail.

## 4   EXPERIMENTS

In this section we verify the benefit of BiSAM across a variety of models, datasets and tasks.

### 4.1   IMAGE CLASSIFICATION ON CIFAR-10/100 AND IMAGENET-1K

We follow the experimental setup of Kwon et al. (2021). We use the CIFAR-10 and CIFAR-100 datasets (Krizhevsky et al., 2009), both consisting of 50 000 training images of size $32 \times 32$, with 10 and 100 classes, respectively. For data augmentation we apply the commonly used random cropping after padding with 4 pixels, horizontal flipping, and normalization using the statistics of the training distribution at both train and test time. We train multiple variants of VGG (Simonyan and Zisserman, 2014), ResNet (He et al., 2016), DenseNet (Huang et al., 2017) and WideResNet (Zagoruyko and Komodakis, 2016) (see Tables 1 and 2 for details) using cross entropy loss. All experiments are conducted on a NVIDIA A100 GPU.

Our two variants of BiSAM are compared against two baselines.

**SGD:** Standard training using stochastic gradient descent (SGD) (see details below)

**SAM:** The original Sharpness-Aware Minimization (SAM) algorithm from Foret et al. (2021)

**BiSAM (tanh):** Algorithm 1 using (17) as the lower bound

**BiSAM (-log):** Algorithm 1 using (18) as the lower bound

The models are trained using stochastic gradient descent (SGD) with a momentum of 0.9 and a weight decay of 0.0005. We used a batch size of 128, and a cosine learning rate schedule that starts at 0.1. The number of epochs is set to 200 for SAM and BiSAM while SGD are given 400 epochs. This is done in order to provide a computational fair comparison as (Bi)SAM uses twice as many gradient computation. Label smoothing with a factor 0.1 is employed for all method. For the SAM and BiSAM hyperparameter $\rho$ we use a value of 0.05. We fix $\mu = 10$ and $\alpha = 0.1$ for BiSAM (tanh) and $\mu = 1.5$ for BiSAM (-log) throughout *all* experiments on both

**Table 1: Test accuracies on CIFAR-10.** BiSAM (-log) has strictly better performance than SAM across *all* models. We include BiSAM (tanh) for completeness which sometimes performs better than BiSAM (-log).

| Model | SGD | SAM | BiSAM (-log) | BiSAM (tanh) |
|---|---|---|---|---|
| DenseNet-121 | $96.14_{\pm 0.09}$ | $96.52_{\pm 0.10}$ | $\mathbf{96.61_{\pm 0.17}}$ | $96.63_{\pm 0.21}$ |
| Resnet-56 | $94.01_{\pm 0.26}$ | $94.09_{\pm 0.26}$ | $\mathbf{94.28_{\pm 0.31}}$ | $94.87_{\pm 0.34}$ |
| VGG19-BN | $94.76_{\pm 0.10}$ | $95.09_{\pm 0.12}$ | $\mathbf{95.22_{\pm 0.13}}$ | $95.01_{\pm 0.06}$ |
| WRN-28-2 | $95.71_{\pm 0.19}$ | $96.00_{\pm 0.10}$ | $\mathbf{96.02_{\pm 0.12}}$ | $95.99_{\pm 0.09}$ |
| WRN-28-10 | $96.77_{\pm 0.21}$ | $97.18_{\pm 0.04}$ | $\mathbf{97.26_{\pm 0.10}}$ | $97.17_{\pm 0.05}$ |
| Average | $95.48_{\pm 0.08}$ | $95.78_{\pm 0.06}$ | $\mathbf{95.88_{\pm 0.08}}$ | $95.93_{\pm 0.08}$ |

**Table 2: Test accuracies on CIFAR-100.** BiSAM (-log) consistently improves over SAM across all models. We include BiSAM (tanh) for completeness which sometimes performs better than BiSAM (-log).

| Model | SGD | SAM | BiSAM (-log) | BiSAM (tanh) |
|---|---|---|---|---|
| DenseNet-121 | $81.31_{\pm 0.38}$ | $82.31_{\pm 0.15}$ | $\mathbf{82.49_{\pm 0.14}}$ | $82.88_{\pm 0.42}$ |
| Resnet-56 | $73.98_{\pm 0.16}$ | $74.38_{\pm 0.37}$ | $\mathbf{74.67_{\pm 0.15}}$ | $74.54_{\pm 0.35}$ |
| VGG19-BN | $74.90_{\pm 0.30}$ | $74.94_{\pm 0.12}$ | $\mathbf{75.25_{\pm 0.24}}$ | $75.12_{\pm 0.34}$ |
| WRN-28-2 | $77.95_{\pm 0.14}$ | $78.09_{\pm 0.13}$ | $\mathbf{78.21_{\pm 0.23}}$ | $78.07_{\pm 0.13}$ |
| WRN-28-10 | $81.50_{\pm 0.48}$ | $82.89_{\pm 0.47}$ | $\mathbf{83.27_{\pm 0.26}}$ | $83.35_{\pm 0.25}$ |
| Average | $77.93_{\pm 0.14}$ | $78.52_{\pm 0.13}$ | $\mathbf{78.78_{\pm 0.09}}$ | $78.79_{\pm 0.14}$ |

CIFAR-10 and CIFAR-100 datasets as a result of a grid search over $\{0.01, 0.1, 1, 10, 100\}$ for $\alpha$ and over $\{0.1, 0.5, 1, 1.5, 2, 4\}$ for $\mu$ using the validation dataset on CIFAR-10 with Resnet-56.

The training data is randomly partitioned into a training set and validation set consisting of 90% and 10%, respectively. We deviate from Foret et al. (2021); Kwon et al. (2021) by using the *validation* set to select the model on which we report the *test* accuracy in order to avoid overfitting on the test set. We report the test accuracy of the model with the highest validation accuracy across the training with mean and standard deviations computed over 6 independent executions. The results can be found in Tables 1 and 2.

For evaluations at a larger scale, we compare the performance of SAM and BiSAM on ImageNet-1K (Russakovsky et al., 2015). We apply each method with $\rho = 0.05$ for both SAM and BiSAM. We use training epochs 90, peak learning rate 0.2, and batch size 512. We employ mSAM (Foret et al., 2021; Behdin et al., 2023) with micro batch size $m = 128$ to accelerate training and improve performance. We set $\mu = 5$ for BiSAM (-log) and $\mu = 20$ and $\alpha = 0.1$ for BiSAM (tanh). Other parameters are the same as CIFAR-10 and CIFAR-100. We run 3 independent experiments for each method and results are shown in Table 3. Note that we do not reproduce experiments of SGD on ImageNet-1K due to computational restriction but it well-documented that SAM and its variants have better performance than SGD (Foret et al., 2021; Kwon et al., 2021; Du et al., 2022).

We find that BiSAM (-log) *consistently* outperforms SGD and SAM across all models on both CIFAR-10 and CIFAR-100, and it also outperforms SAM on ImageNet-1K. In most cases, BiSAM (tanh) has better or almost same performance than SAM. Average accuracies across 5 models of both BiSAM (-log) and BiSAM (tanh) outperform SAM and the result is statistically significant as shown by the small standard deviation when aggregated over all model types. This improvement is achieved *without* modifying the original experimental setup and the hyperparameter involved. Specifically, we use the same $\rho = 0.05$ for BiSAM which has originally been tuned for SAM. The consistent improvement using BiSAM, despite the favorable setting for SAM, shows the benefit of our reformulation based on the 0-1 loss. Note that the generalization improvement provided by BiSAM comes at essentially *no computational overhead* (see Appendix B for detailed discussion).

We recommend using **BiSAM (-log)** as it generally achieves better or comparable test accuracies to BiSAM (tanh). Therefore, we choose BiSAM (-log) as representative while BiSAM (tanh) serves as reference in all tables.

**Table 3:** Test accuracies on ImageNet-1K.

|  | SAM | BiSAM (-log) | BiSAM (tanh) |
|---|---|---|---|
| Top1 | $75.83_{\pm 0.16}$ | $\mathbf{75.96}_{\pm 0.15}$ | $76.02_{\pm 0.08}$ |
| Top5 | $92.47_{\pm 0.02}$ | $\mathbf{92.49}_{\pm 0.10}$ | $92.40_{\pm 0.13}$ |

**Table 4:** Test accuracies for finetuning.

|  | SAM | BiSAM (-log) |
|---|---|---|
| Flowers | $98.79_{\pm 0.07}$ | $\mathbf{98.93}_{\pm 0.15}$ |
| Pets | $93.66_{\pm 0.48}$ | $\mathbf{94.15}_{\pm 0.24}$ |

## 4.2 Finetuning on Oxford Flowers and Pets

We conduct experiments of transfer learning on ViT architectures. In particular, we use pretrained ViT-B/16 checkpoint from Visual Transformers (Wu et al., 2020) and finetune the model on Oxford-flowers (Nilsback and Zisserman, 2008) and Oxford-IITPets (Parkhi et al., 2012) datasets. We use AdamW as base optimizer with no weight decay under a linear learning rate schedule and gradient clipping with global norm 1. We set peak learning rate to $1e-4$ and batch size to 512, and run 500 steps with a warmup step of 100. Note that for Flowers dataset, we choose $\mu = 4$ for BiSAM(-log) and $\mu = 20$ for BiSAM(tanh); and for Pets dataset, set $\mu = 6$ for BiSAM(-log) and $\mu = 20$ for BiSAM(tanh). The results in the table indicate that BiSAM benefits transfer learning.

## 4.3 Incorporation with variants of SAM

Since we just reformulate the perturbation loss of original SAM, existing variants of SAM can be incorporated within BiSAM. The mSAM variant has been combined with BiSAM in experiments on ImageNet-1K. Moreover, we incorporate BiSAM with both Adaptive SAM (Kwon et al., 2021) and Efficient SAM (Du et al., 2022).

**Adaptive BiSAM.** We combine BiSAM with Adaptive Sharpness in ASAM (Kwon et al., 2021) which proposes a normalization operator to realize adaptive sharpness. The Adaptive BiSAM (A-BiSAM) algorithm is specified in detail in Appendix A.1 and results on CIFAR-10 are shown in Table 5. A-BiSAM (-log) *consistently* outperforms ASAM across *all* models on CIFAR-10 except for one on DenseNet-121 where the accuracy is the same.

**Efficient BiSAM.** BiSAM is also compatible with the two ideas constituting ESAM (Du et al., 2022), *Stochastic Weight Perturbation* and *Sharpness-sensitive Data Selection*. A detailed description of the combined algorithm, denoted E-BiSAM, is described in Appendix A.2 and results on CIFAR-10 are shown in Table 6. E-BiSAM (-log) improves the performance of ESAM across *all* models on CIFAR-10.

**Table 5:** Test accuracies of A-(Bi)SAM.

| Model | ASAM | A-BiSAM (-log) |
|---|---|---|
| DenseNet-121 | $96.79_{\pm 0.14}$ | $96.79_{\pm 0.13}$ |
| Resnet-56 | $94.86_{\pm 0.18}$ | $\mathbf{95.09}_{\pm 0.09}$ |
| VGG19-BN | $95.10_{\pm 0.09}$ | $\mathbf{95.14}_{\pm 0.14}$ |
| WRN-28-2 | $96.22_{\pm 0.10}$ | $\mathbf{96.28}_{\pm 0.14}$ |
| WRN-28-10 | $97.37_{\pm 0.07}$ | $\mathbf{97.42}_{\pm 0.09}$ |
| Average | $96.07_{\pm 0.05}$ | $\mathbf{96.14}_{\pm 0.05}$ |

**Table 6:** Test accuracies of E-(Bi)SAM.

| Model | ESAM | E-BiSAM (-log) |
|---|---|---|
| DenseNet-121 | $96.30_{\pm 0.22}$ | $\mathbf{96.35}_{\pm 0.12}$ |
| Resnet-56 | $94.21_{\pm 0.38}$ | $\mathbf{94.60}_{\pm 0.24}$ |
| VGG19-BN | $94.16_{\pm 0.09}$ | $\mathbf{94.43}_{\pm 0.14}$ |
| WRN-28-2 | $95.95_{\pm 0.08}$ | $\mathbf{96.00}_{\pm 0.04}$ |
| WRN-28-10 | $97.17_{\pm 0.09}$ | $\mathbf{97.18}_{\pm 0.05}$ |
| Average | $95.56_{\pm 0.09}$ | $\mathbf{95.71}_{\pm 0.06}$ |

## 5 Related Work

The min-max zero-sum optimization template has been used in recent years in multiple applications beyond SAM (Foret et al., 2021) e.g., in Adversarial Training (Madry et al., 2018; Latorre et al., 2023) or Generative Adversarial Networks (GANs) (Goodfellow et al., 2014).

In particular, the SAM formulation as a two player game that interact via addition, already has precedence in Robust Bayesian Optimization (Bogunovic et al., 2018), where it is called $\epsilon$-perturbation stability. Even though our formulation starts as a zero-sum game (2), a tractable reformulation (15) requires leveraging the bilevel optimization approach (Bard, 2013).

The bilevel paradigm has already seen applications in Machine Learning, in the context of hyperparameter optimization (Domke, 2012; Lorraine et al., 2020; Mackay et al., 2019; Franceschi et al., 2018), meta-learning (Franceschi et al., 2018; Rajeswaran et al., 2019), data denoising by importance learning (Ren et al., 2018), neural architecture search (Liu et al., 2018) and training data poisoning (Mei and Zhu, 2015; Muñoz-González et al., 2017; Huang et al., 2020). Our formulation is the first bilevel formulation in the context of SAM.

ESAM (Du et al., 2022) introduces two tricks, *Stochastic Weight Perturbation (SWP)* and *Sharpness-sensitive Data Selection (SDS)* that subset random variables of the optimization problem, or a subset of the elements in the mini-batch drawn in a given iteration. Neither modification is related to the optimization objective of SAM. Thus, analogous ideas can be used inside our bilevel approach as shown in Algorithm 3. This is useful, as ESAM is able to reduce the computational complexity of SAM while retaining its performance. We can see a similar result when combined with BiSAM.

In ASAM (Kwon et al., 2021), a notion of *Adaptive Sharpness* is introduced, whereby the constraint set of the perturbation $\epsilon$ in (1) is modified to depend on the parameter $w$. This particular choice yields a definition of sharpness that is invariant under transformations that do not change the value of the loss. The arguments in favor of adaptive sharpness hold for arbitrary losses, and hence, adaptivity can also be incorporated within the bilevel formulation of BiSAM as shown in Algorithm 2. Experimental results in Table 5 demonstrate that this incorporation improves performance.

A relationship between the inner-max objective in SAM and a Bayesian variational formulation was revealed by Möllenhoff and Khan (2022). Based on this result, they proposed *Bayesian SAM (bSAM)*, a modification of SAM that can obtain uncertainty estimates. While such results require a continuity condition on the loss c.f. Möllenhoff and Khan (2022, Theorem 1.), their arguments could be applied to any sufficiently tight continuous approximation of the 0-1 loss. Therefore, a similar relationship between our formulation of SAM and the Bayesian perspective could be derived, enabling uncertainty estimates for BiSAM.

In GSAM (Zhuang et al., 2021), propose to minimize the so-called *surrogate gap* $\max_{\epsilon:\|\epsilon\|\leq\rho} L_S(w+\epsilon) - L_S(w)$ and the perturbed loss $\max_{\epsilon:\|\epsilon\|\leq\rho} L_S(w+\epsilon)$ simultaneously, which leads to a modified SAM update. In Liu et al. (2022), it is proposed to add a random initialization before the optimization step that defines the perturbation. In Ni et al. (2022), it is suggested that using the top-k elements of the mini-batch to compute the stochastic gradients is a good alternative to improve the speed of SAM. To different degrees, such SAM variants have analogous versions in our framework.

## 6 Conclusions and Future Work

In this work, we proposed a novel formulation of SAM by utilizing the 0-1 loss for classification tasks. By reformulating SAM as a bilevel optimization problem, we aimed to maximize the lower bound of the 0-1 loss through perturbation. We proposed BiSAM, a scalable first-order optimization method, to effectively solve this bilevel optimization problem. Through experiments on CIFAR-10, CIFAR-100 and ImageNet-1K datasets, BiSAM outperformed SAM meanwhile maintaining a similar computational complexity. In addition, incorporating variants of SAM (*e.g.,* ASAM, ESAM, mSAM) in BiSAM formulation can improve its performance or efficiency further. Moreover, BiSAM also showed good performance in finetuning and noisy label tasks.

Considering the promising results obtained in the context of classification tasks, it is intriguing to investigate the applicability of the BiSAM formulation in the realm of natural language processing (NLP) which would broaden the scope of application. Overall, the insights gained from this work offer new directions and opportunities for advancing the field of sharpness-aware optimization.

**Reproducibility statement.** Our implementation is built upon the PyTorch version of SAM.[1] To reproduce our BiSAM (tanh) and BiSAM (-log) variants, simply modify the perturbation loss function following equations (17) and (18), respectively. The same revision applies when integrating BiSAM with SAM variants such as A-BiSAM (Algorithm 2) and E-BiSAM (Algorithm 3). Implementation details and specific parameters are provided in Section 4 and the supplementary material in Section A.

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

## A  COMPLETE EXPERIMENTS

### A.1  ADAPTIVE BISAM

As we introduced in Section 5, some existing variants of SAM can be incorporated within BiSAM. To demonstrate this, we combine BiSAM with Adaptive Sharpness in ASAM (Kwon et al., 2021). Kwon et al. (2021) propose that the fixed radius of SAM's neighborhoods has a weak correlation with the generalization gap. Therefore, ASAM proposes a normalization operator to realize adaptive sharpness. Following the element-wise operator in Kwon et al. (2021) defined by

$$T_w = \text{diag}(|w_1|, \ldots, |w_b|), \text{ where } w = [w_1, \ldots, w_b], \tag{19}$$

we construct Adaptive BiSAM (A-BiSAM) in Algorithm 2.

---

**Algorithm 2** Adaptive BiSAM (A-BiSAM)

---

**Input:** Initialization $w_0 \in \mathbb{R}^d$, iterations $T$, batch size $b$, step sizes $\{\eta_t\}_{t=0}^{T-1}$, neighborhood size $\rho > 0$, $\mu > 0$, lower bound $\phi$.

1 **for** $t = 0$ **to** $T - 1$ **do**
2      Sample minibatch $\mathcal{B} = \{(x_1, y_1), \ldots, (x_b, y_b)\}$.
3      Compute the (stochastic) gradient of the perturbation loss $Q_{\phi,\mu}(w_t)$ defined in Eq. (14)
4      Compute perturbation $\epsilon_t = \rho \frac{T_w^2 \nabla_w Q(w)}{\|T_w \nabla_w Q(w)\|}$ .
5      Compute gradient $g_t = \nabla_w L_{\mathcal{B}}(w_t + \epsilon_t)$.
6      Update weights $w_{t+1} = w_t - \eta_t g_t$.

---

To compare A-BiSAM with ASAM, we use the same experimental setting as in Section 4.1 except for the choice of $\rho$. For both ASAM and A-BiSAM, we use $\rho = 2$ as a result of a grid search over $\{0.1, 0.5, 1, 2, 3\}$ using the validation dataset on CIFAR-10 with Resnet-56. (Note that we do not use $\rho = 0.5$ as in Kwon et al. (2021) because the results of ASAM with $\rho = 0.5$ cannot outperform SAM in our experiments.) We also have two variants of A-BiSAM compared against ASAM:

**ASAM:** The original Adaptive Sharpness-Aware Minimization (ASAM) algorithm from Kwon et al. (2021)

**A-BiSAM (tanh):** Algorithm 2 using (17) as the lower bound

**A-BiSAM (-log):** Algorithm 2 using (18) as the lower bound

We report the test accuracy of the model with the highest validation accuracy across the training with mean and standard deviations computed over 5 independent executions. The results can be found in Table 7.

**Table 7:** Test accuracies of A-(Bi)SAM on CIFAR-10 dataset.

| Model | ASAM | A-BiSAM (-log) | A-BiSAM (tanh) |
|---|---|---|---|
| DenseNet-121 | $96.79_{\pm 0.14}$ | $96.79_{\pm 0.13}$ | $96.76_{\pm 0.06}$ |
| Resnet-56 | $94.86_{\pm 0.18}$ | $95.09_{\pm 0.09}$ | $94.86_{\pm 0.12}$ |
| VGG19-BN | $95.10_{\pm 0.09}$ | $95.14_{\pm 0.14}$ | $95.19_{\pm 0.15}$ |
| WRN-28-2 | $96.22_{\pm 0.10}$ | $96.28_{\pm 0.14}$ | $96.29_{\pm 0.18}$ |
| WRN-28-10 | $97.37_{\pm 0.07}$ | $97.42_{\pm 0.09}$ | $97.34_{\pm 0.11}$ |

We find that A-BiSAM (-log) *consistently* outperforms ASAM across *all* models on CIFAR-10 except for one on DenseNet-121 where the accuracy is the same. In addition, A-BiSAM (tanh) has same or better performance than ASAM on Resnet-56, VGG19-BN, and WRN-28-2. It may be improved by further tuning $\mu$ and $\alpha$.

## A.2 EFFICIENT BiSAM

A-BiSAM above mainly improves the performance of BiSAM while some variants of SAM can enhance the efficiency like Efficient SAM (ESAM) (Du et al., 2022). As we introduced in Section 5, ESAM proposes two tricks, *Stochastic Weight Perturbation (SWP)* and *Sharpness-sensitive Data Selection (SDS)*, which can also be used in BiSAM. When these two tricks are combined with BiSAM we refer to it as Efficient BiSAM (E-BiSAM) in Algorithm 3.

---

**Algorithm 3** Efficient BiSAM (E-BiSAM)

---

**Input:** Initialization $w_0 \in \mathbb{R}^d$, iterations $T$, batch size $b$, step sizes $\{\eta_t\}_{t=0}^{T-1}$, neighborhood size $\rho > 0$, $\mu > 0$, lower bound $\phi$, SWP hyperparameter $\beta$, SDS hyperparameter $\gamma$.

1 **for** $t = 0$ **to** $T - 1$ **do**
2     Sample minibatch $\mathcal{B} = \{(x_1, y_1), \ldots, (x_b, y_b)\}$.
3     **for** $i = 0$ **to** $d - 1$ **do**
4        **if** $w_t[i]$ *is chosen by probability* $\beta$ **then**
5           $\epsilon_t[i] = \frac{\rho}{1-\beta} \nabla_{w[i]} Q(w_t)$
6        **else**
7           $\epsilon_t[i] = 0$
8     Compute the perturbation loss $Q(w_t + \epsilon_t)$ and construct $\mathcal{B}^+$ with selection ratio $\gamma$:

$$\mathcal{B}^+ = \{(x_i, y_i) \in \mathcal{B} : Q(w_t + \epsilon_t) - Q(w_t) > a\}, \text{ where } a \text{ controls } \gamma = \frac{|\mathcal{B}^+|}{|\mathcal{B}|}$$

9     Compute gradient $g_t = \nabla_w L_{\mathcal{B}^+}(w_t + \epsilon_t)$.
10    Update weights $w_{t+1} = w_t - \eta_t g_t$.

---

To compare E-BiSAM with ESAM, we use the same experimental setting as in Section 4.1. For hyperparameter $\beta$ and $\gamma$ for SWP and SDS respectively, we choose 0.5 for both which is same as Du et al. (2022). We compare two variants of E-BiSAM against ESAM:

**ESAM:** The original Efficient Sharpness-Aware Minimization (ESAM) algorithm from Du et al. (2022)

**E-BiSAM (tanh):** Algorithm 3 using (17) as the lower bound

**E-BiSAM (-log):** Algorithm 3 using (18) as the lower bound

We report the test accuracy of the model with the highest validation accuracy across the training with mean and standard deviations computed over 5 independent executions. The results can be found in Table 8.

**Table 8:** Test accuracies of E-(Bi)SAM on CIFAR-10 dataset.

| Model | ESAM | E-BiSAM (-log) | E-BiSAM (tanh) |
|---|---|---|---|
| DenseNet-121 | $96.30_{\pm0.22}$ | $96.35_{\pm0.12}$ | $96.32_{\pm0.11}$ |
| Resnet-56 | $94.21_{\pm0.38}$ | $94.60_{\pm0.24}$ | $94.32_{\pm0.34}$ |
| VGG19-BN | $94.16_{\pm0.09}$ | $94.43_{\pm0.14}$ | $94.31_{\pm0.12}$ |
| WRN-28-2 | $95.95_{\pm0.08}$ | $96.00_{\pm0.04}$ | $95.95_{\pm0.09}$ |
| WRN-28-10 | $97.17_{\pm0.09}$ | $97.18_{\pm0.05}$ | $97.14_{\pm0.07}$ |

We observe that E-BiSAM (-log) outperformances ESAM across *all* models and E-BiSAM (tanh) has same or better performance except for on WRN-18-10. As a result, E-BiSAM combined with SWP and SDS improves the efficiency of BiSAM meanwhile keeping good performance.

We test on a task outside the i.i.d. setting that the method was designed for. Following Foret et al. (2021) we consider label noise, where a fraction of the labels in the training set are corrupted to another label sampled uniformly at random. Apart from the label perturbation, the experimental setup is otherwise the same as in Section 4.1, except for adjusting $\rho = 0.01$ for SAM and BiSAM when noise rate is 80%, as the original $\rho = 0.05$ causes failure for both methods. We find that BiSAM enjoys similar robustness to label noise as SAM despite not being specifically designed for the setting.

**Table 9:** Test accuracies of ResNet-32 models trained on CIFAR-10 with label noise.

| Noise rate | SGD | SAM | BiSAM (-log) | BiSAM (tanh) |
|---|---|---|---|---|
| 0% | $94.76_{\pm 0.14}$ | $94.95_{\pm 0.13}$ | $\mathbf{94.98_{\pm 0.17}}$ | $95.01_{\pm 0.08}$ |
| 20% | $88.65_{\pm 1.75}$ | $92.57_{\pm 0.24}$ | $\mathbf{92.59_{\pm 0.11}}$ | $92.35_{\pm 0.29}$ |
| 40% | $84.24_{\pm 0.25}$ | $\mathbf{89.03_{\pm 0.09}}$ | $88.71_{\pm 0.23}$ | $88.86_{\pm 0.18}$ |
| 60% | $76.29_{\pm 0.25}$ | $82.77_{\pm 0.29}$ | $\mathbf{82.91_{\pm 0.46}}$ | $82.87_{\pm 0.71}$ |
| 80% | $44.44_{\pm 1.20}$ | $44.68_{\pm 4.01}$ | $\mathbf{50.00_{\pm 1.96}}$ | $48.57_{\pm 0.64}$ |

# B   Computational complexity

We claim that BiSAM has the same computational complexity as SAM. This can be seen from the fact that the only change in BiSAM is the loss function used for the ascent step. By visual inspection of such loss function, its forward pass has the same complexity as that of vanilla SAM: we use the same logits but change the final loss function. Hence, the complexity should remain the same. We report timings of each epoch on CIFAR10 in our experiments. Note that time of training on CIFAR10 and CIFAR100 are roughly same.

**Table 10:** Time of each epoch.

| Model | SAM (cross-entropy) | BiSAM (logsumexp) |
|---|---|---|
| DenseNet-121 | 58s | 64s |
| Resnet-56 | 23s | 30s |
| VGG19-BN | 10s | 16s |
| WRN-28-2 | 19s | 21s |
| WRN-28-10 | 65s | 71s |

The relatively small computational overhead (10% in the best cases) is most likely due to cross entropy being heavily optimized in Pytorch. There is no apparent reason why logsumexp should be slower so we expect that the gap can be made to effectively disappear if logsumexp is given similar attention. In fact, it has been pointed out before that logsumexp in particular is not well-optimized in Pytorch (Bolte, 2020).

To provide further evidence that the computation overhead can be removed, we time the forward/backward of the ascent loss in both Pytorch and TensorFlow with batch size=128 and number of class=100 for 10k repetitions. We find that in tensorflow BiSAM would even enjoy a speedup over SAM.

**Table 11:** Compare Pytorch with TensorFlow.

| Model | SAM (cross-entropy) | BiSAM (logsumexp) |
|---|---|---|
| Pytorch | 2.40s | 3.96s |
| Tensorflow | 3.25s | 2.34s |

