# OpenReview forum: "Improving SAM Requires Rethinking its Optimization Formulation"
_ICLR.cc/2024/Conference — Submitted to ICLR 2024_

### Official Review · Reviewer_6ZVi · 2023-10-31

**Soundness:** 2 fair
**Presentation:** 2 fair
**Contribution:** 2 fair
**Rating:** 3
**Confidence:** 3

**Summary:**

Motivated by the min-max nature of sharpness-aware minimization (SAM), the paper develops a reformulation of SAM using bilevel optimization called BiSAM. This approach can be viewed as a continuous relaxation of SAM with 0-1 loss and the authors argue it provides a tighter bound on generalization gap. Numerical experiments are conducted to demonstrate the empirical improvement of BiSAM over original SAM.

**Strengths:**

The presentation of the work is clear to me. In particular, the motivation of using different bounds for min and max parts to handle the discrete 0-1 loss looks insightful to me and this naturally leads to the development of bilevel approaches. The numerical comparisons of BiSAM and SAM are conducted on various models to demonstrate the possible practical benefits of BiSAM. The authors also show that BiSAM can be easily embedded into variants of SAM such as adaptive SAM and efficient SAM.

**Weaknesses:**

- The significance of the contribution is unclear to me. According to the numerical experiments conducted, the improvement of BiSAM over original SAM seems quite minor. See Questions for more discussion on this point.
- In turn, this makes me wonder if there is any real benefit of approximating 0-1 loss (via this bilevel method) instead of cross-entropy. The authors argue that directly building SAM over 0-1 loss provides tighter generalization bound but more discussions are required.

**Questions:**

Major comments:

- The major concern is the significance of the contribution. When comparing the performance of BiSAM and SAM, it turns out the improvement in test accuracy is quite minor. Based on the standard deviation provided, it is hard to conclude that the improvements are due to randomness or indeed statistically significant.
- The motivation behind BiSAM is the benefit of approximating 0-1 loss via BiSAM over continuous surrogate like cross-entropy. The intuition of lower bounding the max part is convincing but I wonder the actual benefit of doing so. In particular, could the authors justify the using of lower bound (11) provides tighter bound than using $L^{ce}$ in the original SAM? I think illustration by a simple synthetic example can help a lot, or through theoretical derivations.


Other comments:

- Is the term $-\frac{1}{\mu}\log k$ missing in the right-hand-side of Equation (14)?
- In the paragraph below Equation (8), what does this sentence mean: ``it would be wrong to simply replace it by cross-entropy"? I think it is valid to upper bound the right-hand-side of (8) by $L^{ce}$. Is there anything I'm missing?

---

> ### Author Response · Authors · 2023-11-15
> **Response to Reviewer 6ZVi**
>
> We thank the reviewer for their valuable feedback and address all remaining concerns below:
>
> > Q1. The major concern is the significance of the contribution. When comparing the performance of BiSAM and SAM, it turns out the improvement in test accuracy is quite minor. Based on the standard deviation provided, it is hard to conclude that the improvements are due to randomness or indeed statistically significant.
>
> A1. We respectfully disagree regarding the perceived minor improvement between BiSAM and SAM. Comparing with the increased accuracy from SGD to SAM, the average improvement from BiSAM to SAM achieves 33.33% on CIFAR-10 and 44.07% on CIFAR-100.
>
> It is important to highlight the consistency of these enhancements across various settings, which provides high confidence that the improvements are not due to randomness. We effectively have a much larger sample size that reduces the variance as the 'Average' row in the paper indicates. However, we are not just better on average, but on **each configuration**! Our validation is across $3$ tasks, $5$ datasets, $8$ models, and $2$ incorporations with SAM variants (ASAM and ESAM). It would be a very low probability event to do well across all these configurations solely due to noise.
>
> Our evaluation is in-line with several SOTA papers from the SAM literature such as ASAM [Kwon et al., 2021], ESAM [Du et al., 2022] and GSAM [Zhuang et al., 2021]. These papers either provide confidence intervals, that are comparable with ours, or provide no confidence interval -- in both cases they instead rely on showing improved performance across multiple settings.
>
>
> > Q2. The motivation behind BiSAM is the benefit of approximating 0-1 loss via BiSAM over continuous surrogate like cross-entropy. The intuition of lower bounding the max part is convincing but I wonder the actual benefit of doing so. In particular, could the authors justify the using of lower bound (11) provides tighter bound than using $L^{ce}$ in the original SAM? I think illustration by a simple synthetic example can help a lot, or through theoretical derivations.
>
>
> A2. To more convincingly make our argument, we provide the following **new example**.  Consider two possible vector of logits:
>
> - ``Option A``: $(1/K + \delta, 1/K - \delta, \ldots, 1/K)$
> - ``Option B``: $(0.5 - \delta, 0.5 + \delta, 0, 0, \ldots, 0)$
>
> where $\delta$ is a small positive number, and assume the first class is the correct one, so we compute the cross-entropy with respect to the vector $(1, 0, \ldots, 0)$.
>
> For ``Option A``, the cross-entropy is $-\log(1/K + \delta)$, which tends to **infinity** as $K\to \infty$ and $\delta \to 0$.  For ``Option B``, the cross-entropy is $-\log(0.5-\delta)$, which is a small constant number. Hence, an adversary that maximizes an upper bound like the cross-entropy, would always choose ``option A``.  However, note that ``Option A`` **never** leads to a maximizer of the 0-1 loss, since the predicted class is the correct one (zero loss). In contrast, ``Option B`` always achieves the maximum of the 0-1 loss (loss is equal to one), even if it has low (i.e., constant) cross-entropy. This illustrates why maximizing an upper bound like cross-entropy provides a possibly weak weight-perturbation. Unfortunately, this subtlety cannot be summarized in one plot.
>
> If, in contrast, we maximize a lower bound like $\tanh(\max_{j \neq y}z_j - z_y)$ where $z$ is the vector of logits (the quantity inside tanh we call negative margin) we obtain: negative-margin for ``option A`` is $-\delta$ whereas negative-margin for ``option B`` is $2 \delta$, so an adversary maximizing a lower bound would choose ``option B`` which correctly leads to a maximization of the 0-1 loss.
>
> Finally, note that maximizing an upper bound is **never** done in the optimization literature as it doesn't provide any improvement guarantees (this should be clear), in contrast, maximizing a lower bound always ensures improvement and is the reason why, for example, gradient descent converges for Lipschitz-gradient objectives.
>
> > Q3. Is the term $-\frac{1}{\mu}\log k$ missing in the right-hand-side of Equation (14)?
>
> A3. Thanks for pointing this out. We will revise it in the paper.
>
> > Q4. In the paragraph below Equation (8), what does this sentence mean: ``it would be wrong to simply replace it by cross-entropy"? I think it is valid to upper bound the right-hand-side of (8) by $L^{ce}$. Is there anything I'm missing?
>
> A4. We explain ''it would be wrong to simply replace it by cross-entropy'' in the following paragraph in the paper. There, we claim that it is valid to minimize an upper bound of 0-1 loss such as $L^{ce}$. However, there is no guarantees when it comes to maximizing an upper bound. Therefore, in Section 3, we decouple Equation (8) to a bilevel problem to retain $L^{ce}$ for the minimizer while devise a **lower bounds** for the maximizer.

---

> ### Author Response · Authors · 2023-11-20
> **Happy to provide further clarification**
>
> Dear Reviewer 6ZVi,
>
> We appreciate the constructive feedback from the reviewer. During the rebuttal, we have taken the following actions:
>
> - Explained that the improvement from SAM to BiSAM is substantial compared with the increased accuracy from SGD to SAM. The consistent enhancement on each configuration across various settings also provides high confidence to the improvement of BiSAM.
> - Provided an example illustrating the failure of maximizing the upper bound while the lower bound approach succeeds.
> - Added $-\frac{1}{\mu}\log k$ to Equation (14).
>
> We believe that our rebuttal addresses all of the main concerns in your review, and as such, we hope that you’ll consider increasing your score. Otherwise please let us know about any remaining concerns. Thank you!
>
> Best regards,
>
> Authors

---

### Official Review · Reviewer_Z9xG · 2023-10-31

**Soundness:** 2 fair
**Presentation:** 2 fair
**Contribution:** 3 good
**Rating:** 6
**Confidence:** 4

**Summary:**

The article 'Rethinks Sharpness-Aware Minimization' addresses the issue of 0-1 loss being non-continuous and thus not amenable to gradient-based optimization. This is tackled by having the minimizing (maximizing) player use an upper bound (lower bound) surrogate for the 0-1 loss. The problem is analyzed thoroughly and the task is clearly defined.

**Strengths:**

The BiSAM method proposed in the paper somewhat resolves the issue of optimizing the 0-1 loss using gradients.
This method has been validated across multiple datasets, demonstrating its advantages over SAM through extensive experiments.

**Weaknesses:**

1. "The idea of BiSAM is very good, but its performance in experiments is only marginally better than SAM. The improvement over SAM is often within the range of error, making it hard to believe that it is an enhancement of SAM.

2. Can you explain why BiSAM using tanh as the lower bound has higher test accuracy on CIFAR-10 compared to using -log as the lower bound, but the results are the opposite on CIFAR-100?

3. Could you combine the characteristics of tanh and -log to create a new lower bound that is suitable for different datasets?"

**Questions:**

See weakness.

---

> ### Author Response · Authors · 2023-11-16
> **Response to Reviewer Z9xG**
>
> We thank the reviewer for their valuable feedback and address all remaining concerns below:
>
> > Q1. "The idea of BiSAM is very good, but its performance in experiments is only marginally better than SAM. The improvement over SAM is often within the range of error, making it hard to believe that it is an enhancement of SAM.
>
> A1. We appreciate the positive acknowledgment of the idea behind BiSAM. We kindly point out that comparing with the increased accuracy from SGD to SAM, the average improvement from BiSAM to SAM achieves 33.33% on CIFAR-10 and 44.07% on CIFAR-100.
>
> It is important to highlight the consistency of these enhancements across various settings, which provides high confidence to the improvement of BiSAM. We effectively have a much larger sample size that reduces the variance as the 'Average' row in the paper indicates. However, we are not just better on average, but on **each configuration**! Our validation is across $3$ tasks, $5$ datasets, $8$ models, and $2$ incorporations with SAM variants (ASAM and ESAM).
>
> Our evaluation is in-line with several SOTA papers from the SAM literature such as ASAM [Kwon et al., 2021], ESAM [Du et al., 2022] and GSAM [Zhuang et al., 2021]. These papers either provide confidence intervals, that are comparable with ours, or provide no confidence interval -- in both cases they instead rely on showing improved performance across multiple settings.
>
>
>
>
> > Q2. Can you explain why BiSAM using tanh as the lower bound has higher test accuracy on CIFAR-10 compared to using -log as the lower bound, but the results are the opposite on CIFAR-100?
>
> A2. We want to clarify that the statement regarding BiSAM(tanh) having higher accuracy than BiSAM(-log) on CIFAR-10, and the reverse on CIFAR-100, is inaccurate. The table below illustrates the mean accuracy differences, calculated as $\operatorname{BiSAM(-log)} - \operatorname{BiSAM(tanh)}$, demonstrating that BiSAM(-log) outperforms BiSAM(tanh) in most cases. Moreover, in Table 1 and 2, BiSAM(-log) exhibits more stable performance than BiSAM(tanh). Consequently, we recommend the utilization of BiSAM(-log).
> | Model  | CIFAR-10 | CIFAR-100 |
> |:---:|:---:|:---:|
> | DenseNet-121  |  -0.02   |  -0.39  |
> | Resnet-56  |   -0.59    |  **+0.13**    |
> | VGG19-BN  |  **+0.21**    |  **+0.13**   |
> | WRN-28-2  |   **+0.03**    |  **+0.14**    |
> | WRN-28-10  |  **+0.09**    |   -0.08  |
>
>
> > Q3. Could you combine the characteristics of tanh and -log to create a new lower bound that is suitable for different datasets?
>
> A3. We thank the reviewer for this insightful suggestion. We combine 2 perturbation loss functions $Q_{tanh}$ and $Q_{-log}$ of the maximizer, which use Equation (17) and (18) respectively in Equation (14). We define a new mixed loss $Q_{mix}=\max(Q_{tanh},Q_{-log})$ which is a (tighter) lower bound. In experimental results on Resnet-56 shown below, BiSAM (mix) demonstrates a balanced performance between the two original lower bounds. It would be an interesting future direction to construct other lower bounds and understand their properties.
>
> | Model  | SAM | BiSAM (-log) | BiSAM (tanh) | BiSAM (mix) |
> |:---:|:---:|:---:|:---:|:---:|
> | CIFAR-10   |  94.09   |  94.28  | 94.87 | 94.56 |
> | CIFAR-100  |   74.38 |  74.67 | 74.54 | 74.61 |

---

> ### Author Response · Authors · 2023-11-20
> **Happy to provide further clarification**
>
> Dear Reviewer Z9xG,
>
> We appreciate the constructive feedback from the reviewer. During the rebuttal, we have taken the following actions:
>
> - Explained that the improvement from SAM to BiSAM is substantial compared with the increased accuracy from SGD to SAM. The consistent enhancement on each configuration across various settings also provides high confidence to the improvement of BiSAM.
> - Compared BiSAM (-log) and BiSAM (tanh) in detail.
> - Created a new mix loss combining BiSAM (-log) and BiSAM (tanh) and showed its empirical results.
>
>
> We believe that our rebuttal addresses all of the main concerns in your review, and as such, we hope that you’ll consider increasing your score. Otherwise please let us know about any remaining concerns. Thank you!
>
> Best regards,
>
> Authors

---

> > ### Comment · Reviewer_Z9xG · 2023-11-23
> >
> > Appreciate the authors efforts on the rebuttal. The author address most of my question, due to the interesting idea, I will raise the score, however, due to the limited performance gain, raising one point is the maximum.

---

### Official Review · Reviewer_idNJ · 2023-11-01

**Soundness:** 2 fair
**Presentation:** 3 good
**Contribution:** 3 good
**Rating:** 6
**Confidence:** 3

**Summary:**

The authors propose BiSAM, an algorithm for solving the Sharpness Aware Minimization (SAM). The original SAM algorithm (Foret et al., 2021) aims to solve a min-max problem of a differentiable loss that upper bounds the 0-1 loss. The authors suggest that directly considering the min-max problem of the 0-1 loss provides a tighter generalization bound. Accordingly, the BiSAM algorithm aims to solve the problem by substituting the 0-1 loss with a differentiable upper bound for minimization and a differentiable lower bound for maximization. The experimental results demonstrate the benefit of BiSAM compared with the original SAM algorithm.

**Strengths:**

- The idea of directly aiming to solve min-max of 0-1 loss and accordingly minimizing/maximizing different surrogates brings novelty.
- The authors provide theoretically justified lower bound for practical implementation. They also provide a clear discussion on two different choices of surrogates.
- The numerical results demonstrate that BiSAM improves accuracy.

**Weaknesses:**

- The numerical results show limited improvements. Also, in some other works (Foret et al., 2021; Liu et al., 2022), SAM achieves accuracy higher than the accuracy of BiSAM in this paper (with the same model and number of epochs).

Liu, Y., Mai, S., Cheng, M., Chen, X., Hsieh, C. J., & You, Y. (2022). Random sharpness-aware minimization. Advances in Neural Information Processing Systems, 35, 24543-24556.

**Questions:**

1. Please see the weakness section.
2. The original SAM aims to solve $ \min_w \max_{\epsilon: \Vert \epsilon \Vert_2 \leq \rho} L^{\mathrm{ce}}_S (w + \epsilon) + h(\Vert w\Vert^2_2 / \rho)$, whereas BiSAM aims to solve (15).

    Is there any theoretical result that could compare the gaps between $\min_w L^{01}_\mathcal{D} (w)$ and the two different solutions (the aim of SAM and the aim of BiSAM, respectively)?

---

> ### Author Response · Authors · 2023-11-18
> **Response to Reviewer idNJ**
>
> We thank the reviewer for their valuable feedback and address all remaining concerns below:
>
> > Q1. The numerical results show limited improvements. Also, in some other works (Foret et al., 2021; Liu et al., 2022), SAM achieves accuracy higher than the accuracy of BiSAM in this paper (with the same model and number of epochs).
>
> A1. We respectfully disagree regarding the spirit of the limited improvement comment. While the improvement appears limited in absolute value, compared to the increased accuracy from SGD to SAM, the average improvement from BiSAM to SAM achieves 33.33% on CIFAR-10 and 44.07% on CIFAR-100. The consistent enhancements across various settings provide high confidence that the improvements are not due to randomness. In other words, BiSAM are not only better in the aggregate but on **each configuration**! Our validation is across $3$ tasks, $5$ datasets, $8$ models, and $2$ incorporations with SAM variants (ASAM and ESAM).
>
> Regarding the higher SAM accuracy in existing work, we suppose the reviewer is referring to the results of WRN-28-10 on the CIFAR datasets. Note that our experiments reuses the hyperparameters optimized on Resnet-56 (see the experimental section) and is thus potentially suboptimal for WRN-28-10. For this reason we conduct _new_ experiments, in which we follow an existing setup optimized for WRN-28-10 from the ESAM paper [Du et al., 2022]. To be specific, we change $\rho=0.1$, the peak learning rate of $0.05$ and the weight decay of $0.001$ for both SAM and BiSAM. The table below shows results of SAM in [Foret et al., 2021], and SAM and BiSAM (-log) in our setting. Our test accuracy of SAM is even higher than the SAM and RSAM papers, and importantly BiSAM still outperforms SAM.
>
> | Dataset  | SAM [Foret et al., 2021] | SAM (ours) | BiSAM (-log) |
> |:---:|:---:|:---:|:---:
> | CIFAR-10   |  $97.3\pm0.1$   |  $97.31\pm0.09$  | $97.38\pm0.10$ |
> | CIFAR-100  |   $83.5\pm0.2$ | $83.79\pm0.19$   | $83.83\pm0.23$  |
>
> Concerning the existing experiments in the paper, they should be compared against the [ASAM paper](https://arxiv.org/abs/2102.11600) [Kwon et al., 2021], whose experimental setup we follow. We closely match the performance of the ASAM paper with our baselines, SGD, SAM and ASAM, across all models and datasets. _Without changing the setting_, We show that BiSAM improves consistently.
>
> We believe this addresses the concern regarding matching existing literature.
>
> > Q2. The original SAM aims to solve $\min_{w} \max_{\epsilon:\| \epsilon \|_2 \leq \rho} L^{ce}_S(w+\epsilon) + h(\|w\|_2^2/\rho)$, whereas BiSAM aims to solve (15). Is there any theoretical result that could compare the gaps between $\min_w L_D^{01}(w)$ and the two different solutions (the aim of SAM and the aim of BiSAM, respectively)?
>
>
> A2. We can claim that $\min_w L_D^{01}(w)$ for BiSAM is less than or equal to that of SAM. To make a stronger claim it is required to refine the current generalization bound [Thm. 1, Foret et al. 2020] which does not distinquish between random and adversarial noise (c.f. the commentary in [Andriushchenko & Flammarion 2022]). This is a very interesting open problem.
>
> [Andriushchenko & Flammarion. 2022] Maksym Andriushchenko and Nicolas Flammarion. Towards understanding sharpness-aware minimization. In International Conference on Machine Learning (ICML), 2022.

---

> ### Author Response · Authors · 2023-11-21
> **Happy to provide further clarification**
>
> Dear Reviewer idNJ,
>
> We appreciate the constructive feedback from the reviewer. During the rebuttal, we have taken the following actions:
>
> - Explained that the improvement from SAM to BiSAM is substantial compared with the increased accuracy from SGD to SAM. The consistent enhancement on each configuration across various settings also provides high confidence to the improvement of BiSAM.
> - Provided the new experimental results matching existing literature.
>
> We believe that our rebuttal addresses all of the main concerns in your review, and as such, we hope that you’ll consider increasing your score. Otherwise please let us know about any remaining concerns. Thank you!
>
> Best regards,
>
> Authors

---

### Official Review · Reviewer_duHp · 2023-11-01

**Soundness:** 4 excellent
**Presentation:** 4 excellent
**Contribution:** 4 excellent
**Rating:** 10
**Confidence:** 4

**Summary:**

The paper investigates applying SAM to the actual metric (0-1 loss) in the classification problem and proposes reformulating the original zero-sum game in SAM as a bilevel optimization problem. The bilevel optimization formulation is motivated by the relation between cross-entropy and 0-1 loss. To counter the non-differentiability of the 0-1 loss, the proposed method (BiSAM) introduces a surrogate loss coupled with the cross-entropy loss. The empirical results show consistent improvement over SAM and orthogonal improvements when accompanied with other SAM variants (ASAM and ESAM).

**Strengths:**

- The approach is simple, scalable, and theoretical-sound
- The flow is easy to follow
- The improvements are convincing and validated in many learning scenarios, including standard learning, fine-tuning and noisy-data learning

**Weaknesses:**

- As mentioned in the conclusion, it will be great to see if BiSAM benefits other domains, e.g. NLP

**Questions:**

- In A.3, to further ground BiSAM's improvements under noisy labels. Can the authors provide the number with noisy learning approaches, e.g., mixup?

---

> ### Author Response · Authors · 2023-11-16
> **Response to Reviewer duHp**
>
> We sincerely appreciate the thoughtful feedback and the positive evaluation of our paper. We address all remaining concerns below:
>
> > Q1. As mentioned in the conclusion, it will be great to see if BiSAM benefits other domains, e.g. NLP.
>
> To explore the applicability of BiSAM in NLP, we conducted BERT finetuning on the CoLA dataset following [this tutorial](https://mccormickml.com/2019/07/22/BERT-fine-tuning/). We use the same setting in this tutorial and the same hyperparameters as for CIFAR-10 for SAM and BiSAM. The Matthews Correlation Coefficient (MCC) for evaluating models finetuned with SAM and BiSAM are $51.9$ and $56.0$, respectively. Note that MCC of base optimizer AdamW is $49.8$ in comparison. It seems like a promising direction to investigate BiSAM for the NLP domain, which we leave for future work.
>
> > Q2. In A.3, to further ground BiSAM's improvements under noisy labels. Can the authors provide the number with noisy learning approaches, e.g., mixup?
>
> In response to the request, we do a grid search for mixup $\alpha$ values, ranging from {0.01, 0.05, 0.1, 1, 2, 4, 8, 16}, as detailed in [Zhang et al., 2022]. The table below shows the mean best accuracy of 3 runs achieved with mixup for varying noise rates. BiSAM performs better than mixup either in noisy labels task.
>
> | Noise rate  | SGD | mixup | SAM | BiSAM (-log) |
> |:---:|:---:|:---:|:---:|:---:|
> | 0%  | 94.76 | 94.43   | 94.95   |  94.98 |
> | 20%  | 88.65 | 91.66   |  92.57  | 92.59   |
> | 40%  | 84.24 |  87.58  |  89.03  |  88.71   |
> | 60%  | 76.29 | 79.64   |  82.77  |  82.91   |
> | 80%  | 44.44 |  46.17  |  44.68  |  50.00   |
>
> [Zhang et al., 2022] Hongyi Zhang, Moustapha Cisse, Yann N Dauphin, and David Lopez-Paz. mixup: Beyond empirical risk minimization. In ICLR, 2018.

---

> > ### Comment · Reviewer_duHp · 2023-11-21
> > **Response**
> >
> > Appreciate the authors efforts on the rebuttal.
> >
> > While other reviewers are complaining the limited improvements between BiSAM and SAM, I still vote for acceptance for this paper. The idea is very simple and elegant, which brings great application values. Furthermore, the improvements BiSAM provides are ***very consistent*** in almost every setting including different level of noisy labels, different models, different SAM variants, etc.
> > It's a good paper for me.

---

> > > ### Author Response · Authors · 2023-11-21
> > > **Thank you**
> > >
> > > We sincerely appreciate your support and positive sentiment towards our work. We would like to thank the reviewer again for the feedback as well as the active engagement in the rebuttal!

---

### Meta-Review · Area_Chair_xpNB · 2023-12-11

**Metareview:**

This paper studies a version of SAM named BiSAM, which is based on bilevel optimization problem. The bilevel optimization is closely related to the version studied in SAM (except that the perturbation direction is based on the lower bound of 0-1 loss). The paper provides empirical evidence to support the efficacy of the approach.

**Justification For Why Not Higher Score:**

The arguments in the paper lack theoretical rigor and are, at best, very hand-wavy. The paper needs more rigor. This could have been lesser issue if the empirical results were strong but I agree with the reviewers that the improvements on the empirical side are relatively minor. So overall, I recommend rejection in the current form and encourage the authors to provide more theoretical grounding for the approach before publication.

**Justification For Why Not Lower Score:**

N/A

---

### Decision · Program_Chairs · 2024-01-16

Reject